Cytokine and microRNA levels during different periods of paradoxical sleep deprivation and sleep recovery in rats

Brianza-Padilla Malinalli 1
Sánchez-Muñoz Fausto 2
Vázquez-Palacios Gonzalo 3
Huang Fengyang 4
Almanza-Pérez Julio César 5
Bojalil Rafael rafaelbojalil@gmail.com 2 6
Bonilla-Jaime Herlinda bjh@xanum.uam.mx 7
1 Posgrado en Biologia Experimental, División de Ciencias Biológicas y de la Salud, Universidad Autonoma Metropolitana Iztapalapa , Ciudad de Mexico , Mexico
2 Departamento de Inmunología, Instituto Nacional de Cardiologia Ignacio Chavez , Ciudad de Mexico , Mexico
3 Colegio de Ciencias y Humanidades, Universidad Autonoma de la Ciudad de Mexico , Ciudad de Mexico , Mexico
4 Laboratorio de Investigación en Farmacología y Toxicología, Hospital Infantil de Mexico Federico Gomez , Ciudad de Mexico , Mexico
5 Departamento de Ciencias de la Salud, División de Ciencias Biológicas y de la Salud, Universidad Autonoma Metropolitana Iztapalapa , Ciudad de Mexico , Mexico
6 Departamento de Atención a la Salud, Universidad Autónoma Metropolitana Xochimilco , Ciudad de México , México
7 Departamento de Biologia de la Reproducción, División de Ciencias Biológicas y de la Salud, Universidad Autonoma Metropolitana Iztapalapa , Ciudad de Mexico , Mexico
Young Howard
Electronic publication date: 2018 Sep 13
Publication date: 2018
Volume: 6
Electronic Location ID: e5567
Received 2018 Mar 14; Accepted 2018 Aug 13
Copyright: ©2018 Brianza-Padilla et al.
Copyright year: 2018
Copyright holder: Brianza-Padilla et al.
License: This is an open access article distributed under the terms of the Creative Commons Attribution License, which permits unrestricted use, distribution, reproduction and adaptation in any medium and for any purpose provided that it is properly attributed. For attribution, the original author(s), title, publication source (PeerJ) and either DOI or URL of the article must be cited.
License URL: https://creativecommons.org/licenses/by/4.0/

Keywords: MicroRNAs, Sleep deprivation, Cytokines, Sleep recovery, Inflammatory response, mi

Funding: Mexico’s Consejo Nacional de Ciencia y Tecnología #248825/349311 Funding was provided by Mexico’s Consejo Nacional de Ciencia y Tecnología through Grant #248825/349311. The funders had no role in study design, data collection and analysis, decision to publish, or preparation of the manuscript.

==============================
Background

Sleep has a fundamental role in the regulation of homeostasis. The aim of this study was to assess the effect of different periods of paradoxical sleep deprivation (PSD) and recovery on serum levels of cytokines and miRNAs related to inflammatory responses.

Methods

Male Wistar rats were submitted to a PSD of 24, 96, or 192 h, or of 192 h followed by 20 days of recovery (192 h PSD+R). The concentrations of corticosterone, cytokines (IL-6, TNF, IL-10, Adiponectin) and miRNAs (miR-146a, miR-155, miR-223, miR-16, miR-126, miR-21) in serum were evaluated.

Results

At PSD 24 h a significant increase of IL-6 and decrease of IL-10 were observed. At PSD 96h adiponectin increased. At 192 h of PSD IL-6 increased significantly again, accompanied by a threefold increase of IL-10 and an increase of serum corticosterone. After 20 days of recovery (192 h PSD+R) corticosterone, IL-6 and TNF levels increased significantly, while IL-10 decreased also significantly. Regarding the miRNAs at 24 h of PSD serum miR-146a, miR-155, miR-223, and miR-16 levels all increased. At 96 h of PSD miR-223 decreased. At 192 h of PSD decreases in miR-16 and miR-126 were observed. After recovery serum miR-21 increased and miR-16 decreased.

Conclusion

PSD induces a dynamic response likely reflecting the induced cellular stress and manifested as variating hormonal and inflammatory responses. Sleep deprivation disturbed corticosterone, cytokine and miRNA levels in serum related to the duration of sleep deprivation, as short-term PSD produced effects similar to those of an acute inflammatory response and long-term PSD induced long-lasting disturbances of biological mediators.

New & Noteworthy

This is the first study to show that paradoxical sleep deprivation induces distinctive changes in circulating microRNAs related to inflammation and that after 20 days of recovery from 192 h of paradoxical sleep deprivation the inflammatory response is not fully restored.

Introduction

Sleep is a natural, reversible physiological state characterized by a reduction in the response to external stimuli (Jones, 1998). Numerous hypotheses have been posited to explain the function of sleep, but most authors consider it a homeostatic process involved in energy conservation, in nervous system recuperation, thermoregulation, and the immune function (Siegel, 2005). Sleep deprivation constitutes a relevant health problem in modern society (Tobaldini et al., 2017). Research on sleep has paid special attention to the rapid eye-movement (REM) phase or paradoxical sleep because deprivation of this sleep stage has been shown to induce various effects in the brain, some of which are so significant that they can induce morphological changes, that include reduced proliferation of neuronal precursors in the adult hippocampus (Sahu et al., 2013), and modifications of the neurochemical, hormonal and inflammatory mediators (Krueger et al., 2016).

Both at systemic and cerebral levels, cytokines participate as mediators of various complex physiological processes, including sleep regulation (Lorton et al., 2006). Depending on the amount and/or duration of the loss, sleep deprivation produces multiple physiological effects, including elevations in plasma cortisol or corticosterone levels (Suchecki, Tiba & Tufik, 2002), suggesting that PSD induces a stress response (McEwen, 2006). PSD also induces alterations of inflammatory markers such as IL-1β, IL-6, IL-17 and TNF (Irwin, 2006), excessive daytime secretion of IL-6 (Vgontzas et al., 1999), and modifications in coagulation and cell adhesion molecules in humans (Grandner et al., 2013) and in animal models (Yehuda et al., 2009). Inflammation itself can be regulated for example by adiponectin, which induces a reduction in the secretion of the cytokines IL-6 and IL-8 (Dietze-Schroeder et al., 2005), and by microRNAs (miRNAs). Indeed, some miRNAs play a significant role in sleep regulation and may be expressed at distinct moments of sleep and in different brain structures, including cortical areas that regulate sleep duration (Davis, Clinton & Krueger, 2012). One recent study demonstrated that some plasmatic miRNAs can be altered as part of the physiopathology of central hypersomnia (Holm et al., 2014), obstructive sleep apnea (Li et al., 2017) and sleep deprivation (Matos et al., 2014). Six miRNAs (miR-146a, -155, -223, -16, -126 and -21) have been described as potential biomarkers for at least nine non-neoplastic diseases that are associated with important inflammatory processes (Haider et al., 2014). However, to the best of our knowledge, no studies have yet related different periods of PSD and recovery from it to circulating levels of these inflammation-related miRNAs.

Considering that PSD is, in itself, a stressor and that there is scarce information about the effects of different periods of PSD on inflammation-related mediators, and the duration of these alterations, the objectives of the present study were: (1) to determine the effects of different periods of PSD (24 h, 96 h, and 192 h) on serum levels of: (a) corticosterone; (b) IL-6, TNF, IL-10, and adiponectin; and (c) miRNAs related to the inflammatory response: miR-146a, -155, -233, -16, -126 and -21; and (2) if a long-lasting PSD (192 h) has long-term homeostatic consequences, as determined by the same mediators, considering a recovery period of 20 days.

Method

The study utilized 3-month-old male Wistar rats weighing approximately 250 gr that were raised in the animal facilities at Universidad Autónoma Metropolitana -Iztapalapa (UAMI) under the conditions stipulated in the official Mexican norm (NOM-062-ZOO-1999, 2001 revision). The experimental protocol was approved by the Ethics Committee of UAMI (Session 1514, verified 3-15-14-2010-2018). The rats were deprived of paradoxical sleep for 24 h (PSD 24h), 96 h (PSD 96h), and 192 h (PSD 192h); a fourth group was also deprived for 192 h, but measurements were done after 20 days of recovery (192 h PSD+R). These intervals were chosen because many prior studies of sleep deprivation also used a 96 h evaluation period (Machado et al., 2004; Andersen et al., 2005; Machado, Tufik & Suchecki, 2010), but we decided to observe two additional, extreme periods (24 and 192 h) for comparative purposes. The rationale for adding a prolonged recovery period is that this interval may allow the body to recover from acute PSD-induced alterations experienced during 192 h. Given the aims of our study, we considered the 20-day period to acknowledge long-term consequences of long-lasting PSD.

Each group consisted of 5 rats. The control group was housed in normal boxes in the same space at the animals facility. All rats were kept under a light-dark cycle (9 am–9 pm) at a temperature of approximately 24 °C. The method of euthanasia by decapitation is approved by NOM-062-ZOO-1991.

Paradoxical sleep deprivation method

The modified multi-platform method was used to induce PSD (Suchecki & Tufik, 2000). It involves placing the rats in an acrylic box (127 cm long × 44 cm wide × 45 cm high) with a base that has 14 platforms measuring 6.5 cm in diameter × 3 cm high with 10-cm separations. Water is added to the base of the platform to a depth of approximately 1 cm. This method is based on the loss of muscle tone that characterizes the REM sleep condition. When an experimental animal falls into the water, it experiences a sudden loss of the sleep cycle. This REM sleep deprivation method has been studied previously and is considered reproducible (Machado et al., 2004). Studies have demonstrated that using this method of inducing PSD, animals not only lose REM sleep entirely, but also approximately 37% of slow wave sleep (SWS) (Machado et al., 2004).

Although the modified multi-platform method, has been widely used and endorsed for PSD 96 h (Suchecki & Tufik, 2000; Machado et al., 2004), there is no information regardingthe effect of this method with other periods of PSD on loss of REM sleep and non-REM sleep. Thus, in order to acknowledge the brain activity in each of the studied periods of PSD, polysomnography is warranted. Another handicap is that recovery was not studied for shorter periods of PSD.

Quantification of serum corticosterone and cytokines levels

To evaluate corticosterone and cytokine concentrations, sera were obtained by decapitation, ensuring that euthanasia was always carried out at the exact same hour: 9:00 am. Blood was drawn in vacutainer tubes with separation gel, serum samples were aliquoted and preserved at −70 °C until they were used. To measure corticosterone concentrations, an ELISA Test Kit Mouse/Rat was employed (Alpco, Salem, NH, USA) following the manufacturer’s instructions (sensitivity 6.1 ng/ml). To measure cytokine concentrations, determinations were conducted also by ELISA: For IL-6 we used Prepro Tech (Rocky Hill /NJ) (range 62–8,000 pg/mL), for IL-10 and TNF we used Thermo Fisher Scientific kits (Rockford, IL, U.S.A.) with sensitivity <3 pg/ml, and 11.0 pg/ml respectively; and with an Alpco kit (Salem, NH) for adiponectin, with sensitivity of = 0.08 ng/ml.

Serum miRNA quantification

RNA extraction was performed using the miRNeasy Serum/Plasma kit (Qiagen, Valencia, CA, USA). The miRNAs were reverse-transcribed with the TaqMan MicroRNA Reverse Transcription Kit (Applied Biosystems, Thermo Fisher Scientific, Waltham, MA, USA). The RT reaction was incubated for 30 minutes at 16 °C, 30 minutes at 42°C, and 5 minutes at 85 °C. The miRNAs were then amplified using a TaqMan MicroRNA RT kit (Applied Biosystems, Thermo Fisher Scientific), and detected with TaqMan TM MicroRNA Assays hsa/mus miR-146a, miR-155, miR-233, miR-16, miR-126 and miR-21 primers and probes (Applied Biosystems, Thermo Fisher Scientific, Waltham, MA, USA). The cycling conditions were: initial denaturation at 95 °C for 10 min, followed by 45 cycles at 95 °C for 15 s, at 60 °C for 40 s, and at 72 °C for 10 s. PCR was performed with a LightCycler TM 480 II System (Roche Applied Science, Basel, Switzerland) and the LightCycler 480 Probes Master kit (Roche Applied Science). miRNA concentrations were normalized with the concentrations of miR-39. The miRNeasy serum/plasma Spike-In Control was added to samples after the addition of QIAzol lysis reagent (2 µL of 1.6 × 108 copy/µL of cel-miR-39), which can control for varying RNA isolation yields and amplification efficiency. ΔCt values were calculated with problem Ct i.e., the reference Ct at which miR-39 is taken as the reference gene, represented as 2−ΔCt (Dehoux et al., 2003).

Statistical analysis

All serum cytokine concentrations and serum miRNA levels were analyzed by a one-way ANOVA, followed by a post hoc Tukey-Kramer test. NCSS software was used for these analyses. Statistical significance in relation to the control group and between groups was set at p < 0.05.

Results

Corticosterone concentrations increased after long periods of PSD and stayed elevated after recovery

In order to assess the effects of increasing lengths of sleep deprivation on the stress response, corticosterone levels were measured. Short periods of PSD had no significant effect upon serum corticosterone, but it was found sharply elevated after PSD 192 h as compared to the control group [F(4,25) = 17.18,  p < 0.0001]. After recovery (192 h PSD+R) the levels of corticosterone were significantly lower than after PSD 192 h but were still significantly higher than those found in controls (Fig. 1).

Figure 1 Serum corticosterone levels during PSD.

The corticosterone levels increases by effect of PSD at 192 h and 192 h+R. Values represent means ± SEM. ANOVA followed by Tukey’s tests were performed * p < 0.05 compared to control; φp < 0.05 compared to the PSD 192 h+R group.

Circulating cytokines in rats were modified by deprivation of paradoxical sleep for different periods and were maintained altered after recovery

The effects of different periods of sleep deprivation on serum pro- and anti-inflammatory cytokines levels were measured. With the exception of PSD 96 h the levels of IL-6 were significantly higher than in controls in all PSD groups, including those measured after 192 h PSD+R [F(4,25) = 17.39,  p < 0.0001] (Fig. 2A). PSD did not modify TNF concentrations as compared to controls and a significant increment was only seen after recovery, at 192 h PSD+R, [F(4,25) = 8.717,  p = 0.0003] (Fig. 2B). IL-10 levels were significantly lower than in controls at PSD 24, raised significantly after PSD 192 h and dropped again below controls levels after PSD+R [F(4,25) = 32.84,  p < 0.0001] (Fig. 2C). Finally, circulating levels of adiponectin showed a significant increase only after PSD 96 h [F(4,25) = 4.741,  p = 0.0074] (Fig. 2D).

Figure 2 Effect of different periods of PSD on IL-6 (A), TNF (B), IL-10 (C), and adiponectin (D) concentrations.

Values represent means ± SEM. ANOVA followed by Tukey’s tests were performed; * p < 0.05 compared to the control group; φp < 0.05 compared to PSD 192 h+R. (n = 5 for each group).

Changes in miRNA levels were found after the different periods of PSD and recovery

Serum levels of miRNA related to the inflammatory response were measured at the different periods of PSD and after recovery. At PSD 24 h three miRNA were found significantly increased compared to controls: miR-146a [F(4,25) = 9.538,  p = 0.0002] (Fig. 3A), miR-155 [F(4,25) = 5.895,  p = 0.0027] (Fig. 3B), and miR-223 [F(4,25) = 21.75,  p < 0.0001]. The latter was observed decreased at PSD 96 h (Fig. 3C), as well as miR-16, which was found also decreased at PSD 192 h and PSD192+R compared to controls [F(4,25) = 23.29,  p < 0.0001] (Fig. 3D). miR-126 was decreased at PSD 192 h [F(4,25) = 3.669,  p = 0.0213] (Fig. 3E), and miR-21, showed no significant changes during PSD, but an increase was observed at 192 h PSD+R compared to controls and to PSD 192 h [F(4,25) = 15.48,  p < 0.0001] (Fig. 3F).

Figure 3 miRNAs levels in the sera of rats after different periods of PSD and recovery: miR-146a (A), miR-155 (B), miR-223 (C), miR-16 (D), miR-126 (E), and miR-21 (F).

Values represent means ± SEM. An ANOVA followed by a Tukey’s test was performed. *p < 0.05 compared to the control group, and φp < 0.05 compared to the 192 h PSD+R group.

Discussion

We found that different periods of PSD modify corticosterone, IL-6, IL-10, and adiponectin levels, depending on the duration of PSD. Most of these alterations in cytokines did not return to basal values even after 20 days of recovery: only adiponectin did. TNF, which was not modified during the different PSD periods, was found elevated only after recovery. Related to miRNAs, except for miR-21 which was increased only during recovery, the levels of all the other inflammation-related miRNAs (miR-146a, miR-155, miR-223, miR16, miR-126) were also modified by PSD. Notably, differentiating the miRNAs response to that of the cytokines, all these miRNAs but miR-16 which persisted in decreased levels, had returned to base-line levels after the recovery period. These findings suggest that PSD induces a dynamic response pattern that reflects cellular stress that is manifested as hormonal and inflammatory responses in different stages, that could be long-lasting.

REM sleep deprivation may induce a stress response (McEwen, 2006) that involves not only behavioral responses, but is also associated with major modifications in brain biochemistry and in the endocrine and immune systems, that may be dependent on the duration of REM sleep deprivation in a response related to acute or chronic stress. After PSD 24 h, elevated levels of the inflammation-related molecules IL-6 and miR-146a, miR-155 and miR-223 along with a decrease in the levels ofhe anti-inflammatory cytokine IL-10 were observed. Increased expression of miR-146a has been previously reported in the cerebral cortex of rats that were sleep-deprived for 24 h (Matos et al., 2014). Particularly, since miR-146a is upregulated as NF-kB is activated and controls cytokine signaling (Taganov et al., 2006), this increase suggests that a regulatory response is induced by short-term PSD (that is further lost after longer periods of PSD). Also, it seems that the increase of these miRNAs at 24 h may be part of a rapid response to oxidative stress and endotoxemia induced by sleep deprivation (Hirotsu et al., 2013) processes that foster the secretion of proinflammatory cytokines, as in the case of IL-6 (Matos et al., 2014; Novotny et al., 2008). Furthermore, during acute stress a polymorphic region related to the gene of the 5HT transporter modulates the response of inflammatory cytokines (Yamakawa et al., 2015).

During the “resistance phase” of stress, organisms manifest a process of adaptation to those demands during a defined time, and activate a series of mechanisms in a stress response that aims to restore homeostasis (Goldstein & Kopin, 2007) which is what apparently happens when PSD-induced stress is prolonged. During 96 h of PSD, only miR-223, miR-16 and adiponectin were modified, where the level of miR-223 and miR-16 decreased, but adiponectin increased. These changes may be due to a compensatory response that seeks to limit tissue damage or dysfunction related to both the metabolic and inflammatory alterations involved in this phenomenon (Villarreal-Molina & Antuna-Puente, 2012).

After a chronic sleep restriction (192 h) we found increased serum levels of corticosterone, IL-6, and IL-10. The stress and anti-inflammatory cytokines responses seem to be at their peaks. An increase in the levels of IL-10 after long-term sleep deprivation (18 h/day for 21 days) has been previously reported (Venancio & Suchecki, 2015). The increase in IL-6 after PSD 192 h may be part of this anti-inflammatory responses, since IL-6 has been described to play both pro-inflammatory and anti-inflammatory roles because in some scenarios it stimulates IL-10 synthesis (Petersen & Pedersen, 2006), and inhibits TNF and IL-1 production (Wolkow et al., 2015). At this point of PSD, it seems that an effort to closely regulate inflammation is in process in order to control its negative effects and the impairments associated with it in terms of performance, cognition, sleepiness (Yirmiya & Goshen, 2011) and the incidence of degenerative disorders such as diabetes (Petersen & Pedersen, 2006).

Interestingly, at this point miRNAs seem not to be participating in an active control of inflammation, but rather may reflect an exhaustion situation due to the long-term, sustained wakefulness. At PSD 192 h we found that serum miR-146a, miR-155, and miR-223 had returned to base-line levels. However, low levels of miR-16 and miR-126 were observed. In other scenarios—for example, under hyperglycemic conditions—these two miRNAs participate in the suppression of inflammation and ROS production (Harris et al., 2008; Tang et al., 2017). Also, decreased serum expression of miR-126 (Chen et al., 2016) and miR-16 in the brain (Li et al., 2010) has been associated with type II diabetes. Other reports also indicate that miR-16 may play a role in the active coping mechanisms for stress resilience (Zurawek et al., 2017). This miRNA is highly-expressed in serotoninergic (5-HT) and noradrenergic neurons (NA) (Hansen & Obrietan, 2013), both of which show high activity during wakefulness, and enhance their functioning during sleep deprivation (Hipólide et al., 2005).

Reestablishing homeostatic processes is important for organisms, as it induces a period of recuperation and repair after subjection to a stressful condition. It is interesting to note that even after a 20-day recovery period the consequences of a chronic sleep deprivation were present as alterations in biological mediators. Serum IL-10 and miR-16 were found decreased, while corticosterone, IL-6, TNF, and miR-21 levels were increased in the 192 h PSD+R group compared to controls. A chronic sleep deprivation is undoubtedly a stressful situation. Thus, it is plausible that one possible mediator of this delayed return to a homeostatic state would be corticosterone since high levels were found after chronic PSD and even after a recovery period of 20 days. This situation adds to the reported alteration in glucose metabolism (Brianza-Padilla et al., 2016) after PSD. A previous study of the effects of stress hormones on SD showed that some miRNAs can be regulated by corticosterone in tissue under experimental conditions (Mongrain et al., 2010).

Even when the reported recovery period was clearly not enough to a full-return to homeostasis, there are hints to suggest the presence of a repairing response. For instance, since miR-21 has been associated with the protection of various organs from suffering lesions (Tong et al., 2015; Xu et al., 2017), the elevated levels of miR-21 seen after the 20-day recovery period posterior to 192 h of PSD may well be reflecting a mechanism aimed to restore functionality.

Conclusion

Sleep deprivation may disrupt cytokine and miRNA production, seemingly related to stress and inflammatory responses. Changes in cytokines and miRNAs in sera seem to be related to the duration of sleep deprivation, as short-term PSD produced effects similar to those of an acute inflammatory response. Although an anti-inflammatory response may appear as PSD duration increases, the disruption of homeostasis probably leads to long-lasting variations of some set points that may be involved in the development and/or progression of several diseases.

Supplemental Information

Data S1 Each data point indicates the average performance of 5 animals in each evaluated variable

Click here for additional data file.

We thank the Laboratorio Divisional de Biología Molecular de Ciencias Biológicas y de la Salud (UAM-I) for authorizing the use of the specialized equipment required for this study.

Additional Information and Declarations

Competing Interests

Author Contributions

Animal Ethics

Data Availability

The authors declare there are no competing interests.

Malinalli Brianza-Padilla performed the experiments, analyzed the data, prepared figures and/or tables.

Fausto Sanchez-Muñoz conceived and designed the experiments, performed the experiments, analyzed the data, contributed reagents/materials/analysis tools, authored or reviewed drafts of the paper.

Gonzalo Vazquez-Palacios conceived and designed the experiments, authored or reviewed drafts of the paper.

Fengyang Huang and Julio Cesar Almanza-Perez performed the experiments, analyzed the data.

Rafael Bojalil authored or reviewed drafts of the paper.

Herlinda Bonilla-Jaime conceived and designed the experiments, contributed reagents/materials/analysis tools, authored or reviewed drafts of the paper, approved the final draft.

The following information was supplied relating to ethical approvals (i.e., approving body and any reference numbers):

The Universidad Autonoma Metropolitana Ethical committee provided approval to carry out the study within its facilities (Session 1514, verified 3-15-14-2010-2018).

The following information was supplied regarding data availability:

The raw data are provided in Data S1.

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
