# Peer review of "Cytokine and microRNA levels during different periods of paradoxical sleep deprivation and sleep recovery in rats"

_PeerJ, doi:10.7717/peerj.5567_

## Round 0.1 · original submission · Minor Revisions

Dear Dr. Herlinda

I am pleased to inform you that the reviewers found your paper to be of interest although they have requested minor revisions. In your revised manuscript, please be sure to address each concern in your cover letter so that the reviewers can determine if your revisions address their concerns. Thank you for submitting to PeerJ and we look forward to receiving your revised manuscript.

Reviewer 1 ·

Basic reporting

The manuscript entitled “Cytokine and microRNA concentrations during different periods of paradoxical sleep deprivation and sleep recovery rats” by Herlinda et. al. has profiled serum cytokines and microRNAs at different length of PSD and after a recovery period. Overall the manuscript is well written with clear background knowledge and discussions of the results. Several previous studies reported either the similar cytokine or microRNA profiles in the PSD animal serum. In this respect, lack of novelty is a major concern of this manuscript. However, this manuscript also focused on the recovery period of PSD with certain extend but not with systemic recovery analysis. Systemic analysis of 192h PSD recovery, for example REM rebound profile and another mid-recovery point analysis, will significantly improve the quality of this study. At the present stage of the manuscript, I suggest considering for publication in PeerJ if a revision address the following concerns.

Experimental design

1. What is the status of REM rebound at 20-day PSD animals? Polysomnography of the 192h post PSD recovering animals should help to address this.
2. I suggest another mid-recovery point analysis of the serum parameters.
3. Is 20 days enough to reestablished sleep cycle homeostasis? Please explain with proper citations.
4. Line 260-265; Adiponectin positively regulate IL10 secretion from macrophages (Kumada, 2004). At 96h of PSD there is significant increase in adiponectin but no elevation of IL-10. Do any of these elevated microRNA suppress the early phase (24h, 96h PSD) of IL-10 induction? I suggest performing a met analysis of IL-10 3’UTR for possible microRNA binding, specifically miR-146a/155/223/16. Similarly, for miR-21 binding sites on the 3’UTRs of the serum protein level tested.

Validity of the findings

5. I suggest using conclusion of the result as heading of the result sections.
6. Serum level of IL-6 (Figure1a) is not matching with the data provided in the supplementary file. Please clarify.
7. The result section should be rewritten with clear rational, experimental design and result of each results.
8. Please represent serum corticosterone levels as graph instead of table.
9. I suggest using another independent microRNA reference gene for the serum level microRNA quantification. If the miR-39 was a spike in control, please clearly mention in the materials method section with spiked in copy number.
10. Please mention concentration unit of the cytokines in the supplementary data file.

Additional comments

I enjoyed reviewing this manuscript. Hopefully the suggestions will help to improve the manuscript.

Reviewer 2 ·

Basic reporting

The English language should be improved in some places. Beginning with the title, the use of the word “concentration” implies the microRNAs were quantitatively measured, but the data are qRT-PCR. Therefore, a more appropriate word choice would be “levels.” Also within the title, the word “in” should be placed before “rats.”

Conclusion paragraph within the overview (lines 56-60) has several typos and grammatical issues. Final sentence in this section is a bit vague.

More recent citations should be provided in the introduction and discussion. For example, references in lines 76, 78, 93, 255. The introductory background information and contextualization within the discussion should reflect the most recent knowledge.

Line 78, could the authors elaborate on what they mean by “recently-acquired knowledge”?

Within the results, awkward phrasing can be found on line 187 (“major”).

The figure labels (axes and sample IDs) are very tiny and should be increased in size, as well as the significance symbols within the figures.

Experimental design

The inclusion of multiple time points in the study is a strength, as it provides a big picture view of early, late and recovery phases. The methods are very well described.

Validity of the findings

The methods of the study are well-justified by previous literature.

The discussion is very long – it is rather disproportional to the amount of data presented and almost seems to function as a literature review rather than a succinct interpretation of the data. Part of the issue is over speculation about the meaning of the data. Specific areas include lines 249-252, 260-265, 285-289, 297-302, 307-312, 326-333. The authors should condense these discussions or remove them completely.

The final paragraph of the discussion undercuts the rigorous nature of the study the authors have performed. The limitations mentioned should be moved to the methods section where the description of the modified multi-platform method is first presented. The results section should end on line 338.

Additional comments

The study is rigorously performed, well controlled, and the methods are clearly described. The only short coming is the discussion where the main findings are lost amongst over speculation and extraneous literature references.

---

## Round 0.2 · accepted · Accept

The changes made are satisfactory and the manuscript is now acceptable.

#